

# Post-emergence seedling damage due to vertebrate pests and its impact on soybean establishment

Jay Ram Lamichhane

INRAE, Université Fédérale de Toulouse, UMR 1248 AGIR, Castanet-Tolosan, France

## ABSTRACT

The quality of field crop establishment is an indicator of the productivity and yield quality of a given crop. Several biotic and abiotic factors, as well as cropping practices, affect the quality of field crop establishment. More specifically to soybean, recent studies quantified pre-emergence seedling losses and identified the associated causes of non-emergence. However, little is known about post-emergence seedling damage, mainly due to vertebrate pests, which represent an important problem for growers. A 2-year field observation was conducted to quantify near- and post-emergence seedling damage due to vertebrate pests. The common wood pigeon (*Columba palumbus*) and the European hare (*Lepus europaeus*) were associated with this kind of damage. The characteristic damage due to the common wood pigeon consisted of either partially-damaged cotyledons during emergence or completely uprooted seedlings at emergence. In contrast, damage due to the European hare consisted of chewed seedling or seedling parts. There was significant effect of year ($p < 0.001$) on the final rates of post-emergence seedling damage due to the wood pigeon but not on those due to the European hare. The final rates of post-emergence damage due to the wood pigeon were higher (32% for 2018 and 22% for 2020) compared with those owing to the European hare (18% for 2018 and 17% for 2020). The severity of damage due to vertebrate pests was related to the type of seedling damage that, in turn, affected the capacity of soybean to compensate for post-emergence seedling damage.

## INTRODUCTION

Soybean (*Glycine max* (L.) Merr.), represents one of the most important leguminous crops worldwide, which is mainly grown for the quality of its protein both for food and feed purposes (*Masuda et al., 2009*). In the European Union (EU), an important decrease in soybean acreage has been observed in the last two decades, leading to several socio-economic and environmental concerns (*Labalette et al., 2010*). The EU produces only a small fraction of soybean (2.6 Mt that is equivalent to <0.5% of the global production) with over 85% of the grain consumed in the EU being imported from the American continent (*FAOSTAT, 2018*). More specifically to France, there is a huge gap between the quantity of soybean produced (<0.5 Mt) and consumed (>5 Mt) annually (*FOP, 2018*). To fill this gap, public policy in the EU aims at increasing the soybean acreage with three-fold

Corresponding author
Jay Ram Lamichhane,
jay-ram.lamichhane@inrae.fr

objectives: reducing dependence on genetically-modified soybean imported for feed purpose, satisfying the demand for locally produced non-genetically modified soybean for human consumption, and contributing toward the transition to more sustainable agricultural systems (*Berschneider, 2016*).

Crop establishment consists of three sub-phases: seed germination, seedling emergence, and early seedling growth (*Aubertot et al., 2020*). Recent studies conducted in France reported that several biotic and abiotic stresses can affect the quality of soybean establishment (*Lamichhane et al., 2020a*; *Lamichhane et al., 2020b*). In particular, mechanical obstacles such as soil aggregates and soil surface crusts have been reported to cause seedling emergence losses while soil-borne pests and pathogens were not a major cause of non-emergence. Nevertheless, these studies did not focus on evaluating post-emergence seedling losses due to biotic stresses in general and those due to vertebrate pests in particular, which were frequently observed across the experimental sites. Post-emergence seedling losses due to vertebrate pests represent an important problem for many field crops, leading to severe economic losses across the globe (*Firake, Behere & Chandra, 2016*; *Giunchi et al., 2012*; *McKee et al., 2020*; *Nasu & Matsuda, 1976*).

The objective of this study was to quantify post-emergence seedling damage of soybean due to vertebrate pests at the scale of an experimental site with different spatial replications. This is important to understand whether and to what extent this damage can affect the quality of soybean establishment across the study site. This information may provide important insight for future research that aims to develop and implement cropping practices that limit access to these pests into soybean fields, as well as to design cropping systems that are less vulnerable to vertebrate pest attacks. Collectively, all of this will improve the quality of crop establishment and the competitiveness of the crop in the cropping system.

## MATERIALS AND METHODS

### Desciption of the study site and experimental design

Field observations were carried out for two years (2018 and 2020) in Auzeville experimental station of INRAE (Institut national de recherche pour l'agriculture, l'alimentation et l'environnement; 43.53°N, 1.58°E), southwest of France. The observations were conducted across different field plots of the same study site for the two years. For each year, three field plots grown with the same soybean cultivar (ES Pallador) were chosen to represent spatial replication with regard to the distance from the residential and non-agricultural areas (Fig. 1). The latter areas are characterized by the Midi canal with a small strip of bushes, trees and hedgerow that represent a roosting zone for vertebrate pests. Plot location is an important feature as it may affect the type and abundance of vertebrate pests, and associated post-emergence seedling damage (*Sausse & Lévy, 2020*).

Detailed information on the experimental plots are reported in Table 1. The size of the experimental plots ranged from 0.50 and 2.40 ha and the sowing date was mid-May as generally practiced by growers in the region. The plots were sown with untreated soybean seeds at three cm sowing depth, and 40 seeds m$^{-2}$ planting density with 50 cm inter-row

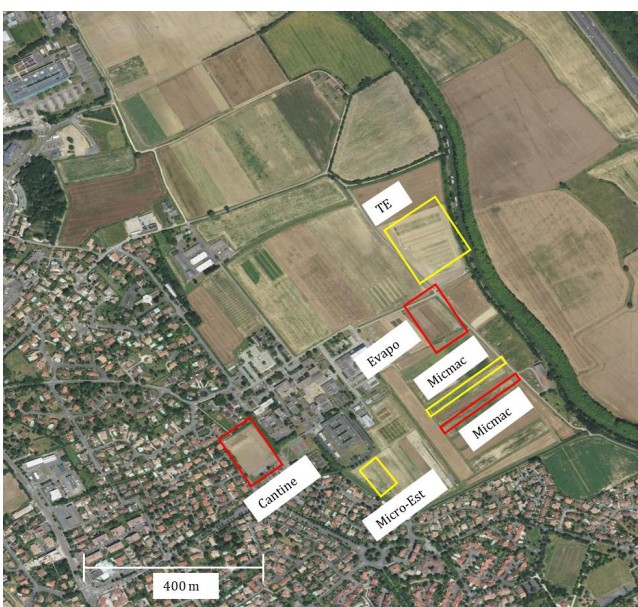

**Figure 1** **An aerial view of the experimental site.** The areas delimited in red and yellow represent the location of the field plots entirely sown with soybean, in 2018 and 2020, respectively. The information reported next to the delimited areas represent the historical names of the plots assigned to facilitate experimental setup (see Table 1). Although the size of the field plots grown to soybean was different within and over the two years, the area used for the counting of post-emergence damage was the same (see Fig. 2). Field plots were located between the residential area and the Midi canal (green strip on the right that is the tree hedge along the canal). A small strip of bushes, trees and hedgerow are located along the Midi canal that represent a roosting area for vertebrate pests. At the level of the experimental site, in both years, neighbouring soybean plots were sown with commonly grown winter-sown (soft and durum wheat, pea, faba bean) and spring-sown (sunflower, maize, sorgum) field crops in the region. ©IGN, 2020.

distance, which corresponds to standard practices of farmers in the region (*Lamichhane et al., 2020a*; *Lamichhane et al., 2020b*). None of the experimental plots were protected by fences or nylon nets to allow free access to vertebrate pests.

The design used for this observational study at the spatial replication scale (i.e., a field plot) is presented in Fig. 2. Each spatial replication included four microplots (25 × 3 m), and each microplot contained four diagonal rows (each 2 linear meters). The countings were performed on 16 rows/plot that represented replicates.

## Field observations and countings

The rate of seedling emergence was determined by the ratio between the number of emerged seedlings at VE stage (i.e., when cotyledon are above the soil surface; *Fehr & Caviness (1977)* and the sowing density. At each observation, the total number of emerged seedlings with or without damage was counted. For damaged seedlings, the type of damage (i.e., partial-damage of cotyledons, total damage due to uprooting of seedlings, or chewed seedlings or seedling parts) and the associated vertebrate pest(s) were identified by direct preliminary field observations and the literature reports (*Firake, Behere & Chandra, 2016*). Preliminary surveys were based on several years of field observations by technicians who

**Table 1  Key description of the field plots used for the observational study.**

| Year | Replicated plots | Plot size (ha) | Distance from the Midi canal (km)[1] | Distance from the residential area (km) | Soil texture (proportion of clay : silt : sand in %) |
|------|------------------|----------------|--------------------------------------|-----------------------------------------|------------------------------------------------------|
| 2018 | Evapo            | 1.09           | 0.10                                 | 0.45                                    | 24: 24: 52                                           |
|      | Micmac           | 0.98           | 0.08                                 | 0.10                                    | 28: 35: 37                                           |
|      | Cantine          | 0.95           | 0.66                                 | 0.02                                    | 25: 38: 37                                           |
| 2020 | TE               | 2.40           | 0.01                                 | 0.55                                    | 37 : 38 : 25                                         |
|      | Micmac           | 0.98           | 0.08                                 | 0.10                                    | 31 : 38 : 31                                         |
|      | Micro-Est        | 0.50           | 0.48                                 | 0.02                                    | 28 : 30 : 42                                         |

**Notes.**

[1] A small strip, composed of bushes, trees and hedgerow up to 30 m high, separates the field plots from the Midi canal that represent a roosting area for vertebrate pests; green highlight: plots that were closer to the canal and farther from the residential area; yellow highlight: plots that were closer to the residential area and farther from the Midi canal.

regularly monitored soybean plots every week at the crop establishment phase. This information allowed us to rule out any post-emergence damage due to other types of vertebrates than those reported here. The countings of post-emergence damage were performed when cotyledons were nearly at the VE stage (i.e., when cotyledons begin to pierce the seedbed surface), and continued across the VC (cotyledon stage where unifoliate leaves are unrolled sufficiently so that the leaf edges are not touching) and V1 (first node stage with fully developed leaves are present at unifoliate nodes) growth stages; *Fehr & Caviness (1977)*. These stages were chosen as post-emergence damage due to vertebrate pests mainly occur at these phases (*Firake, Behere & Chandra, 2016*). This was further confirmed by preliminary observations during which important seedling losses due to vertebrate pests were observed. The countings of post-emergence seedling losses were made every two days for aproximately two weeks. The countings continued until reaching a plateau with no new damaged seedlings for three consecutive countings.

## Statistical analyses

The rate of seedling emergence was determined by using the following formula:

$$Emergence\ (\%) = 100 \times \frac{TES}{SD}$$

Where TES is the total number of emerged seedlings and SD is the sowing density.

The rate of post-emergence damage was calculated using the following formula:

$$Post-emergence\ damage\ (\%) = 100 \times \frac{TDSdt}{TES}$$

Where TDSdt is the number of damaged seedlings per damage type (i.e. partially-damaged cotyledons, uprooted seedlings, or chewed seedlings or seedling parts).

The counted number of plants was transformed in percentage, and the data did not meet the normality assumption. Therefore, a non-parametric Kruskall-Wallis test was applied to test for differences between years. The analysis was performed using the R software (*Hothorn & Everitt, 2009*).
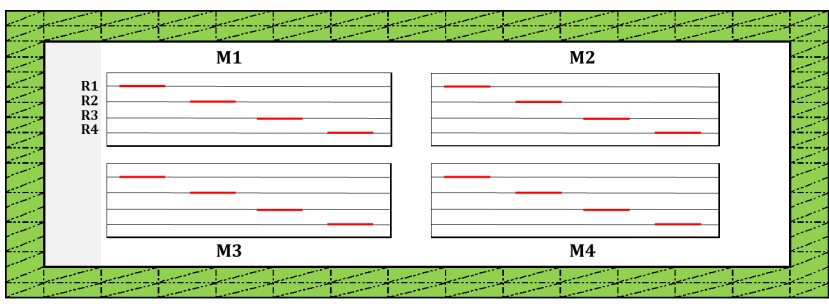

**Figure 2** **Study design used for the observational study at the scale of spatial replication (i.e., plot level) for countings of post-emergence seedling damage due to vertebrate pests.** Following band sowing, each field plot was divided into four micro plots (M1 to M4) by delimiting them with plastic pegs as described previously (*Lamichhane et al., 2020a*). In each microplot, 4 rows in diagonal (R1 to R4), each 2 linear meters/row (i.e., 1m2/row; red lines along each row) were further delimited, (i.e., a total of 4 m2/microplot and 16 m2/plot). For each replicate, this was the area considered for the countings of seedling emergence and post-emergence damage due to vertebrate pests. Two meters in width of the field margin (traced lines filled in green) were excluded from seedling countings to avoid the effect of plot margin on the measured variables. None of the experimental plots were protected by fences or nylon nets to allow free access to vertebrate pests.

## RESULTS

### Soybean emergence rates

Final emergence rates of soybean over the two years are presented in Table 2. The emergence rates were $82 \pm 12$ and $72 \pm 13$ for 2018 and 2020, respectively. There was significant effect of year ($p < 0.001$) on the final rates of seedling emergence.

### Vertebrate pests and characteristics of post-emergence seedling damage

The common wood pigeon, *Columba palumbus* (Linnaeus), and the European hare *Lepus europaeus* (Pallas), were two key vertebrate pests regularly observed in our experimental plots. None of these pests caused pre-emergence damage to soybean seedlings. The post-emergence damage caused by these pests could be recognized based on the characteristics of the damage observed in the field. Seedling damage due to common wood pigeon consisted of either partial damage on the cotyledons as soon as they began to pierce from the seedbed surface (Fig. 3A) or uprooting of seedlings from the seedbed (Fig. 3B). No seedling damage due to common wood pigeon was observed after the VC stage. In contrast, seedling damage due to wild hares was observed between the VC and V1 stage, which consisted of chewed seedlings (Figs. 3C, 3D). Unlike common wood pigeon, wild hares did not cause any uprooting of seedlings.

### Severity of post-emergence seedling damage due to vertebrate pests

The final rates of post-emergence seedling damage, based on individual characteristics observed in the field, are presented in Table 3. There was significant effect of year ($p < 0.001$) on the post-emergence seedling damage due to the wood pigeon (i.e., partially-damaged cotyledons and uprooted seedlings). In contrast, no statistically significant differences

**Table 2** Average final emergence rate (±standard deviation) of soybean (cv. ES Pallador) in relation to spatio-temporal replication of field plots.

| Year | Plot | Emergence (% ± SD) |
|---|---|---|
| 2018 | Evapo | |
| | Micmac | $82^b \pm 12$ |
| | Cantine | |
| 2020 | TE | |
| | Micmac | $72^a \pm 13$ |
| | Micro-Est | |

**Notes.**
Means followed by the different letter are significantly different at $p < 0.05$.

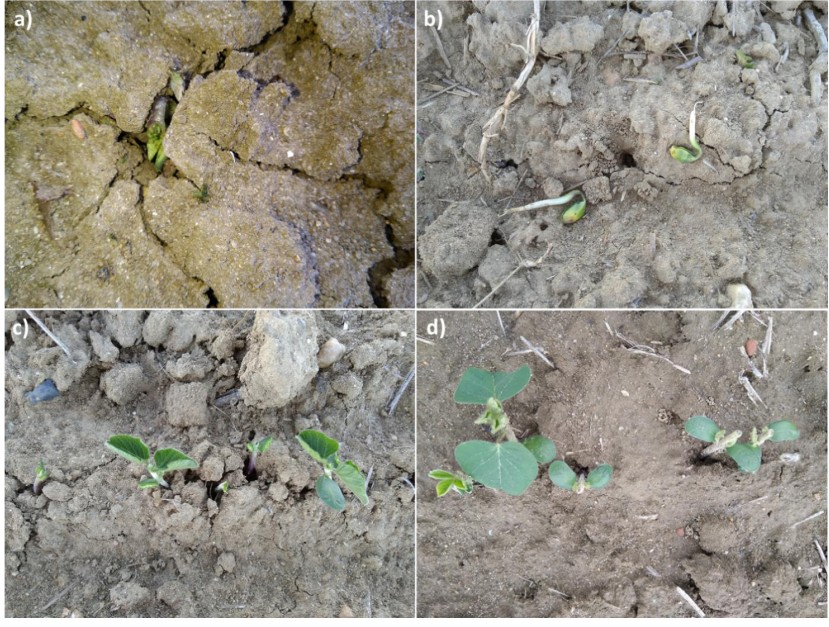

**Figure 3** **Characteristic post-emergence seedling damages caused by the common wood pigeon (A, B) and the European hare (C, D) on soybean.** The cotyledons can be partially damaged by birds during the emergence phase (A) or the entire seedling can be uprooted and scattered on the seedbed surface (B). The damage is reversible to some extent in the first case while it is irreversible in the second case. Following emergence, seedlings can be damaged by wild hares that chew seedlings or seedling parts without uprooting them. Damaged seedlings due to wild hares recover their growth provided that the damage is occurred above the first node although this may have an impact on crop productivity.

($p > 0.05$) were found in the post-emergence seedling damage due to the European hare between the two years. Collectively, the final rates of post-emergence seedling damage, due to both vertebrate pests, significantly differed ($p < 0.001$) between the two years. The rates of post-emergence damage were similar over the two years and they ranged from 8 to 12%, 14 to 20%, and 17 to 18% for partially-damaged cotyledons, uprooted seedlings and chewed seedlings, respectively. The final rates of post-emergence damage were 50% and 39% in 2018 and 2020, respectively.

**Table 3** Post emergence damage (±standard deviation) caused by the common wood pigeon and the European hare on soybean.

| Year | Partially-damaged cotyledons (%) | Uprooted seedlings (%) | Chewed plantlets (%) | Total damage (%) |
|------|----------------------------------|------------------------|----------------------|------------------|
| 2018 | 12 ± 4 | 20 ± 9 | 18 ± 14 | 50 ± 15 |
| 2020 | 8 ± 3 | 14 ± 7 | 17 ± 10 | 39 ± 16 |
| $p$ | *** | *** | NS | *** |

Notes.
NS, not significant.
***$p < 0.001$.

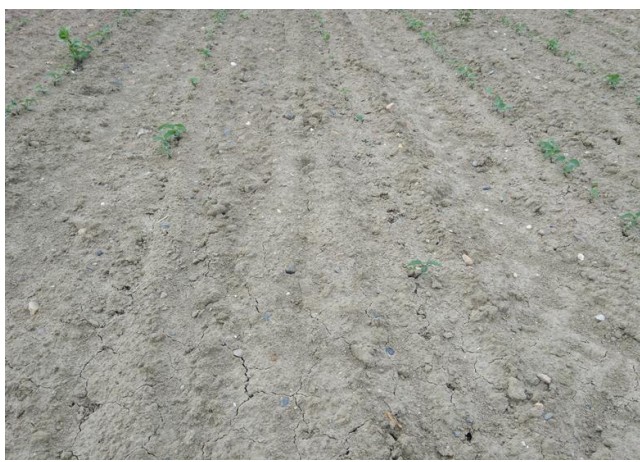

**Figure 4** Post-emergence seedling damage due to the common wood pigeon can lead to total crop establishment failure with severe direct (re-sowing costs, yield losses) and indirect (increased weed pressure during the cropping season with increased weed seedbank for the following years) economic consequences for farmers.

### Recovery ability of damaged seedlings

The capacity of damaged seedlings to recover was dependent on the type and intensity of the initial damage caused by vertebrate pests. No seedling growth recovery occurred leading to severe damage and crop emergence failure when cotyledons were entirely damaged during emergence (i.e., when cotyledons still touch the seedbed surface) or completely uprooted by common wood pigeon (Fig. 4). In contrast, seedlings recovered their growth by developing new leaves beginning from the nodes, when cotyledons were partially damaged at the near VE stage or when seedlings were partially damaged above the first node at the VE to V1 stage (Figs. 5A, 5B).

## DISCUSSION

To the best of the author's knowledge, this is the first study in France and Europe that attempts at quantifying post-emergence damage of soybean due to vertebrate pests via field observations and countings. Previous studies reported crop damage due to vertebrate pests, with millions of dollars of crop losses (*Elser et al., 2019*; *Gebhardt et al., 2011*; *Swanepoel et al., 2017*). However, these losses were mainly related to advanced stages of the crop cycle (i.e., the near harvesting phase) including grain or fruit feeding. In contrast, only
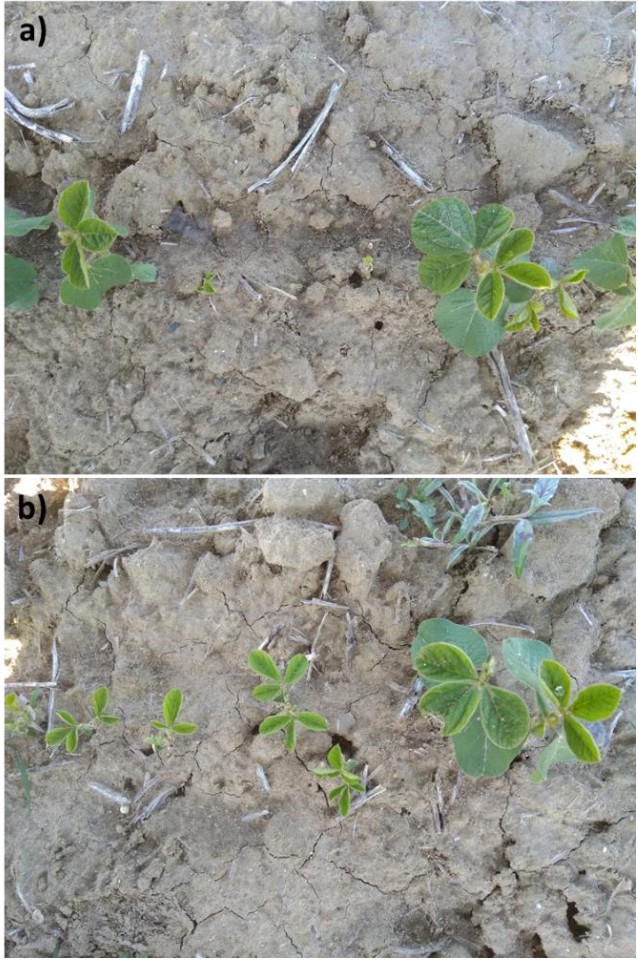

**Figure 5** **Heterogeneous growth stage of soybean seedlings due to post-emergence seedling damage by vertebrate pests.** Damaged seedlings partly compensate their growth that mainly depends on the pest type and growth stage during which the damage occurred.

little is known on the impact of birds and wild animals on the very early stage of field crops (i.e., seed germination, seedling emergence, and early development stages) that can jeoparadize the quality of field crop establishment. A few studies assessed seed and seedling damage that occurred before the crop establishment phase on corn (*Furlan et al., 2017*; *Khan, Javed & Zeeshan, 2015*; *Wise, 2018*), wheat (*Kennedy & Connery, 2008*; *Khan, Javed & Zeeshan, 2015*), oilseed rape (*Schillinger & Werner, 2016*), speciality crops (*Werner et al., 2015*), and soybean (*Firake, Behere & Chandra, 2016*).

A previous study (*Firake, Behere & Chandra, 2016*) on soybean showed that the rate of seed and seedling damage due to Colombidae is strictly dependent on the crop growth stage. Indeed, the authors highlighted that severe damage occurred at the seed germination, seedling emergence, and green cotyledon stages, followed by only moderate to low damage occurred at the V1 stage, and almost negligeble damage after this stage. Nevertheless, there was no pre-emergence damage due to Colombidae across our study site and that

the initial damage commenced to appear nearly at the VC stage, when cotelodons started to pierce the seedbed surface. This difference could be due to many factors including the bird species causing damage, geographical region of the study, the type of landscape, the distance of the experimental site from the residential area, sowing dates, and the type of seedbed preparation. In the study by *Firake, Behere & Chandra (2016)*, the spotted dove, *Spilopelia chinensis* (Scopoli), and occasionally feral pigeon, *Columba livia* (Gmelin), were associated with heavy damage in newly sown soybean fields in northeast India. In contrast, the common wood pigeon and the European hare were associated with post-emergence damage in this study that was carried out in southwestern France and with much earlier sowing dates (i.e., mid-May). This might have affected the bird types involved in seedling damage.

In addition to the common wood pigeons, post-emergence seedling damage due to the European hares was observed during this study. A recent research from central India (*Bayani et al., 2016*) reported damage to soybean and many other crops due to wild mammalian herbivores, including hares, although the damage regarded much advanced stages of the crop. In general, the type of wild mammalian herbivores and their associated damage to crops depend also on the landscape around the field including field-edge habitat and distance from residential area (*Sellers et al., 2018*).

Here, post-emergence seedling damage due to the common wood pigeons was distinguished from that due to the European hares based on the type of damage and crop growth stage. Indeed, the former caused seedling damage at earilier stage than do the latter and the damage type was generally distingushable. Nevertheless, there were a few cases of mixed post-emergence seedling damage caused by both vertebrate pests on the same seedlings (e.g., partially damaged cotyledons recover their growth but they are later attacked by the European hares), which were difficult to distinguish. Future studies are needed to make wider observations of different vertebrate pests on different crop species and the associated seedling damage, which may help precisely estimate seedling damage due to each specific pest cropwise.

In this study, the temporal (i.e., year) effect was significant on the post-emergence damage due to the common wood pigeon but not on that due to the European hare. This contrast is interesting and might be due to the capacity of these two pests to move at different scales. For example, the common wood pigeon, as many other birds, is very mobile animal, capable of flying at a large scale within a short period of time. Consequently, the post-emergence damage of a given crop due to birds is determined by three spatial scales (*Sausse & Lévy, 2020*): (i) the regional scale where they decide to settle for the season (nesting aimed at causing post-emergence damage) that will determine a pool of birds causing post-emergence damage; (ii) the landscape scale, where they select the places to feed as part of their daily activity; and (iii) the plot scale where they look for food items. In contrast to birds, the movement of the European hare remains limited within a restricted area that may explain the similar level of post-emergence damage occurred between the two years of study. Future studies that take into account a wider spatio-temporal replication of field plots are needed to better understand in this regard.

Previous studies reported seed and seedling damages due to Corvidae on many field crops including corn and wheat (*Kennedy & Connery, 2008*; *Khan, Javed & Zeeshan, 2015*; *Wise, 2018*). While Corvidae were occasionally observed across the experimental site, no important damage due to them was found on soybean seed or seedlings. Nevertheless, these birds were more commonly found across neighbouring field plots sown with corn and sunflower. It is possible that Corvidae damage on soybean was negligible because these birds prefer sunflower or corn rather than soybean. However, this study did not investigate the potential correlation between the crop species grown in neighbouring plots and the type and abundance of bird species that needs further exploration.

Post-emergence seedling damage due to vertebrate pests can cause severe yield losses. However, the estimation of these losses is extreamly difficult for several reasons. First, the correlation between post-emergence seedling damage due to vertebrate pests and subsequent yield losses are not necessarily linear (*Brown et al., 2007*). This is especially true for crops that can compensate emergence losses and seedling damage due to their tillering (e.g., wheat) and branching capacity as well as indeterminate or semi-determinate growth (e.g., soybean). In contrast, the correlation between post-emergence seedling damage and subsequent yield losses could be linear for those crops that do not have any compensation ability following early stage damage, such as maize, sugar beet or sunflower. Therefore, as suggested previously (*Swanepoel et al., 2017*), the impact of any stress and the level of damage at the seedling stage should be viewed as damage and not necessarily as crop losses for the first group of crops. In contrast, this level of damage may correspond to subsequent crop losses for crops without any compensation ability. This means that, seedling damage for the second group of crops can be proposed as a good indicator of crop losses. A previous study reported that for temperate crops, 10% seedling losses resulted in 9% crop losses at harvest (*Myllymaki, 1989*). This study showed that the level of compensation of soybean to seedling damage depends on the seedling growth stage and the type of the damage itself. When cotyledons were completely damaged before the development of first nodes (i.e., before the V1 stage), seedlings most often were unable to recover. In contrast, when seedlings were partially damaged at the V1 stage and above the first node, they could develop new leaves beginning from the first nodes which, however, led to a very heterogeneous growth stage of the crop.

## CONCLUSIONS

The present study provides a first estimation of seedling damage due to the common wood pigeon and the European hare that are key vertebrate pests affecting nearly emerged and emerged soybean seedlings in southwestern France. Although the rates of soybean seedling damage due to vertebrate pests were relatively high, most often, the crop was able to partially compensate this damage through its branching capacity and semi-determinate or indeterminate growth habit. However, in some cases, post-emergence damage may directly affect soybean yield especially when cotyledons are entirely damaged during emergence (i.e., when cotyledons still touch the seedbed surface) or completely uprooted by birds.

The results presented in this study were based on 2-year field observations at a small-scale spatial replication. The lack of a large-scale spatial replication and the number of crop

genotype (only one) considered represent a limit of this study. The heterogeneous size of the plots used as spatial replication consititutes another drawback of this study. Indeed, the size of a plot determines the available edge habitat that, in turn, may affect the extent and type of post-emergence damage. Future studies that take into account different study sites and crop genotypes, heterogeneous landscape composition, and sowing dates are needed to provide insights into potential correlations between the quality of soybean establishment and grain yield, both in terms of quality and quanity. Finally, the database generated by this study can be used to feed a qualitative model called CESIM (Crop Establishment SIMulator; *Lamichhane et al., 2020c*). This model, in addition to emergence losses due to abiotic and biotic factors, and cropping practices, also takes into account the impact of post-emergence seedling damage on the quality of field crop establishment.

## ACKNOWLEDGEMENTS

The author thanks: Didier Raffaillac for his kind availability and feedback provided on the research topic during field observations, and l'UMT PACTOLE for supporting the initiative to promote the results presented in this paper.

### Funding
The author received no funding for this work.

### Competing Interests
The author declares that he has no competing interests.

### Author Contributions
- Jay Ram Lamichhane conceived and designed the experiments, performed the experiments, analyzed the data, prepared figures and/or tables, authored or reviewed drafts of the paper, and approved the final draft.

### Data Availability
   Data are available in the Supplemental Files.

### Supplemental Information
Supplemental information for this article can be found online at http://dx.doi.org/10.7717/peerj.11106#supplemental-information.

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
