# Peer review of "Post-emergence seedling damage due to vertebrate pests and its impact on soybean establishment"

_PeerJ, doi:10.7717/peerj.11106_

## Round 0.1 · original submission · Major Revisions

Thank you for your manuscript. It was reviewed by three experts who all agreed that revisions are needed. Most of the points raised by the reviewers are focused on the overall organization of the manuscript; however, reviewers 1 and 2 pointed out some missing information in the methods that need to be addressed. Reviewer 1 would like more detail on how the authors positively identified damage. Reviewer 2 would like much more detail on the specifics of the site used.

Please respond thoroughly to the three reviews and address the issues they have raised. All three reviewers enjoyed your manuscript and I look forward to the revised copy.

·

Basic reporting

This manuscript is generally well-written, and the introduction clearly shows the context for the paper and shows the need for this research. However, in some places editing for correct English language would be helpful. There are several minor misspellings (e.g. “sobean” on line 38, “majour” on line 49, “assesed” on line 170, “thier” on line 282, etc). There are also some cases of awkward wording (e.g. line 105, “was occurred” on line 171, etc).

Tables and Figures:
Table 1: It is unclear in the table (and also from the methods section) what exactly the F and p values refer to. Does this p value indicate a significant difference in emergence by year as indicated on line 124, or does it have to do with a difference by distance to the canal/residential areas?

Figures 3, 4, and 5 are very useful in visualizing the damage described in the paper.

Experimental design

The author clearly defined their research question, and I am convinced that this research is important and will be useful to inform future studies. I was unclear, however, about some aspects of the study design. In some cases, this may just require further explanation in the methods section.
1. Lines 94-97 describe damage characteristic to hares and pigeons. Do you have a citation to support this, or did you do direct observations of vertebrates damaging plants? If so, this should be described in the methods. How can you rule out damage by other types of vertebrates (such as Corvidae mentioned on lines 226-232)?
2. More description of the study site is needed. I cannot determine what type of habitat immediately surrounds each of the study plots. Figure 1 makes it seem that each plot is embedded within a larger agricultural field. Was the entire area planted in soybeans, or just the small plots? In addition, the size of experimental plots (line 78) seems quite small. Is this typical of the area? The size of the plot determines the amount of edge habitat, and edge effects may influence the amount and type of damage you see per plot.
3. The terminology used in line 84 is misleading as this seems to be a purely observational study. Perhaps the authors mean “study design” or something similar instead of “experimental device?”
4. Lines 117-119 need further clarification. Do you mean that you transformed the data in order to fit a normal distribution? What exactly did you test with the ANOVA? Just differences in damage between years?
5. Lines 138-140 describe the effect of distance to residential area on damage, but the data are not shown. Table 1 further lists the distances to residential areas and canal for each plot. It is unclear why the data referred to in lines 138-140 are not shown when they seem to be relevant to the study. If the authors do not wish to discuss this part of the study, I suggest they remove mention of the distance effect from both the results and discussion.

Validity of the findings

Assuming the authors can justify their characterization of damage due to hares and pigeons, I found the conclusions to be well supported and explained. The authors are clear about the relatively limited scale of this study and the need for future research, but their findings represent a useful contribution to the field of study.

Additional comments

I enjoyed reading this manuscript, and found it to be fairly clear and concise. My main concerns, as stated above, are related to the justification and description of the study design. In particular, I believe the author should spend much more time explaining how certain types of damage were linked to pigeons vs hares, as this is the main premise of the study.

·

Basic reporting

no comments, see general comments

Experimental design

no comment

Validity of the findings

The observations are limited to one site and two years only and no particularly significant outputs or crop practices which may reduce vertebrate pest damage are supplied.

Additional comments

The manuscript supplies new information about soybean damage by two major Vertebrate Pests: the common wood pigeon (Columba palumbus) and the European hare (Lepus europaeus). The observations are limited to one site and two years only and no particularly significant outputs or crop practices which may reduce vertebrate pest damage are supplied. The reader cannot understand if the reported significant damage rates resulted in any impact on final crop stand and on yield, in terms of quantity and quality.
Material and Methods and other sections need improvements. Nevertheless information supplied represents a contribution to vertebrate pest risk assessment in soybean; at least if data will be analyzed alongside many more future data.
Therefore manuscript publication is encouraged
M&M
A complete description of material and methods should be given. Any method you use in a scientific research has to be objective, effective and replicable by anyone; therefore, all the details needed to evaluate and replicate the experiment have to be given.
All the agronomic characteristics should be given:
- Soybean variety
- Sowing date
- Seed density
- Soil texture of sampling site – variety, ...)
- …….....

At least a short description of plants in the woody strip (bushes, trees, hedgerow nearby Midi canal) should be given (e,g. main plant species, were they high?,...)

A specific table completely separated by results should be prepared.

Current Table 1 is mixing M&M information with results and this should be avoided.

In order to improve this, I would suggest to have a Table 1 associated to M&M, with coordinates of sampling sites and the main agronomic characteristics; a second Table will give the results.

Another important factor to be considered is the presence of alternative food sources for wild fauna; other soybean fields.

L 117-118 If data didn’t meet the Normality assumption a non-parametric approach should be applied

RESULTS

L 122: The author reports information related to M&M (Cv ES Pallador); please move this information to M&M as above suggested;
L 139-140 This was the most interesting factor to show (compared to year not so interesting), why didn't the author provide at least a graph with distance from naturalistic or residential areas plotted to damage?

Discussion

L 168 – 164: it is truethat little is known on the impact of birds and wild animals on the very early stage of field crops but a little bit more than reported is known in Europe ; long term studies on maize damage at early stages have shown that bird/wild animal damage was low; therefore some references about maize should be added;
L 198-200 unjustified statement; please add data demonstrating the statement; there are few significant outputs of this work: why did not you present data to justify this factor effect? The reader cannot actually understand; please add relevant analyzes;
L 221-222 unjustified statement; please add reference
L 231 Negligible not negligeble
L 261 . 262: unjustified statement; please supply data demonstrating this
In order to exploit experimental results it would be useful to demonstrate:
- the effect of plant damage reported in Table 1 on plant mortality and final plant density (crop stand);
- the effect of vertebrate pest damage at early stages on yield.

Conclusion

The word IPM has never been used in the text!! The author should add something about implications of the study results for IPM implementation (solution to contrast vertebrate pest damages in soybean)

L 268 which is the insight for future research? No spatial factor demonstrated or different effect on crop yield depending on damaged crop stage.

L273-274: Why didn't the author give an example of the output of the model using the data base of the present work?

References
Some references does not allow to find the cited publication (e.g. Wise, K., 2018. Open Field Study with “Avipel Shield” Seed Treatment on Field Corn to Deter Birds from Feeding on Corn Seed and Corn Seedlings

Table 1: Damage by wood pigeon and hare should be identified in the columns.

Reviewer 3 ·

Basic reporting

The manuscript entitled “Post-emergence seedling damage due to vertebrate pests and its impact on soybean establishment” is well written with clear and professional English. Introduction and background of the study are adequately provided in the manuscript and relevant prior literature are appropriately referenced

Experimental design

Methods are described with sufficient information to be reproducible by another investigator. However, certain parts need to be explained in detail (See comments).

Validity of the findings

Results and conclusions are well stated and sufficiently discussed. The manuscript can be accepted after minor revision

Additional comments

Line 17: Mention the name of the authority after each scientific names at lease first time in abstract and on main text (Also add the same in Line 126)
“The common wood Pigeon (Columba palumbus…………) and the European hare (Lepus europaeus…………..)”

Line 49: change ‘majour’ to word ‘major’
Line 62 add space between ‘and2020’
Line 185-186: the sentence need to be modified as “the spotted dove, Spilopelia chinensis (Scopoli) and feral pigeon, Columba livia Gmelin……”

Line 187: Delete the sentence “and with late sowing dates”. It seems meaningless here.

Line 193: Mention the Scientific name of “European hares”.
Methodology of damage assessment in case of European hares is not clear. How the damage was assessed? How many times in a day? How did author know that the damage was due to European hares?

Line 200: (data not shown): Without data, how can author make such sentences?

---

## Round 0.2 · Minor Revisions

Thank you for making the required changes to the manuscript. Both reviewers agree that you have addressed their major concerns. They have some additional (very minor) points to address. Please read through their comments, make the changes, and send the manuscript back for a final look.

·

Basic reporting

I Think this is an overall improved manuscript, and the author made the suggested changes well. There are still a few (mainly small) issues to be addressed, as listed below.

• Line 28, Add the word “for” after “compensate”
• Line 33, make “crop” plural
• Line 35, make “purpose” plural

• In the new manuscript, the author clarifies on lines 103-105 that they used preliminary observations to identify types of damage by pigeons vs hares. However, they deleted the description of the types of damage typically caused by each, which should still be included in this paper. I see these descriptions are now in the results section. Since these are being reported as results, it seems like the authors should provide more information in the methods about their preliminary observations (e.g. how long did they watch for, did they ever see other wildlife in the field that didn’t damage the crops, etc). Alternatively, the authors could move the descriptions of types of damage to the methods section only, and then explain that they used their knowledge of the type of damage caused by pigeons vs. hares to categorize the damage data they collected and reported in this experiment.

• On line 133 and elsewhere, the author refers to damage due to birds. Earlier you claim that all damage was specifically due to the Common wood pigeon, so please be more specific here and replace “birds” with pigeon or Common wood pigeon. Otherwise it seems that this could be caused by other bird species, which would be an important distinction.

• The figure captions are cut off in my version of the PDF so I cannot read them to tell if they have been changed.

• I understand the author’s reluctance to address the effect of the distance to residential area on damage due to the small sample size, and I commend them for including these data in the supplementary table. However, the reader needs to understand this limitation as well, especially since the author has still included a statement on lines 141-145 about their findings about the distance effect. I think the author needs to explain to readers that these are just observed patterns from the data, but that the sample size is too small to conduct a robust statistical test. This could potentially even go in the methods section under the statistical analysis subheading, so it is presented early on and the author is explicit about it. Alternatively, the author could opt to remove these findings from both the results and discussion section.

Experimental design

see above

Validity of the findings

see above

Additional comments

see above

·

Basic reporting

manuscript OK

Experimental design

manuscript OK

Validity of the findings

OK

Additional comments

The author properly addressed the comments. The manuscript can be accepted after a further minir revision:

Please replace

Dimitri, G., Yuri, V., Albores-Barajas, N., Emilio, B., Lorenzo, V., Cecilia, S., 2012. Feral Pigeons: Problems, Dynamics and Control Methods, Integrated PestManagement and Pest Control Current and Future Tactics Dr. Sonia Soloneski(Ed.), ISBN: 978-953-51-0050-8, InTech,.

with

Giunchi, D., Albores-Barajas Y.V, Baldaccini N.E., Vanni, L., Soldatini, C., S., 2012. Feral Pigeons: Problems, Dynamics and Control Methods, Integrated PestManagement and Pest Control Current and Future Tactics Dr. Sonia Soloneski(Ed.), ISBN: 978-953-51-301 0050-8, InTech

---

## Round 0.3 · accepted · Accept

Thank you for your efforts. I am happy to accept your manuscript.